# A Nationwide Survey and Risk Assessment of Ethyl Carbamate Exposure Due to Daily Intake of Alcoholic Beverages in the Chinese General Population

**DOI:** 10.3390/foods12163129

**Published:** 2023-08-21

**Authors:** Zifei Wang, Pengfeng Qu, Yunfeng Zhao, Yongning Wu, Bing Lyu, Hongjian Miao

**Affiliations:** China National Center for Food Safety Risk Assessment, NHC Key Laboratory of Food Safety Risk Assessment, Chinese Academy of Medical Science Research Unit (2019RU014), Beijing 100021, China; wangzifei@cfsa.net.cn (Z.W.); qupengfeng@cfsa.net.cn (P.Q.); zhaoyf@cfsa.net.cn (Y.Z.); wuyongning@cfsa.net.cn (Y.W.); lvbing@cfsa.net.cn (B.L.)

**Keywords:** ethyl carbamate, total diet study, GC-MS, margin of exposure, virtually safe dose, T25, lifetime cancer risk

## Abstract

Ethyl carbamate (EC) is carcinogenic, and, in China, oral intake of EC mainly occurs as a result of the consumption of alcoholic beverages. To obtain the latest EC intake and risk analysis results for the general population in China, the China National Center for Food Safety Risk Assessment (CFSA) conducted the sixth total diet study (TDS) as a platform to analyze EC contents and exposure due to the intake of alcoholic beverages. A total of 100 sites in 24 provinces were involved in the collection and preparation of alcohol mixture samples for the sixth TDS. There were 261 different types of alcohol collected across the country, based on local dietary menus and consumption survey results. Ultimately, each province prepared a mixed sample by mixing their respective samples according to the percentage of local consumption. The EC levels of these twenty-four mixed samples were determined using our well-validated gas chromatography-mass spectrometry (GC-MS) method. The values ranged from 1.0 μg/kg to 33.8 μg/kg, with 10.1 μg/kg being the mean. China’s EC daily intake ranged from 0.001 ng/kg bw/d to 24.56 ng/kg bw/d, with a mean of 3.23 ng/kg bw/d. According to the margin of exposure (MOE), virtually safe dose (VSD), and T25 risk assessments of the carcinogenicity of EC, the mean lifetime cancer risk for the Chinese population was 9.8 × 10^4^, 1.5 × 10^−7^, and 8.6 × 10^−8^, respectively. These data show that the carcinogenicity of EC in the general Chinese population due to alcoholic intake is essentially minimal.

## 1. Introduction

The toxicological mode of action of ethyl carbamate (EC), a substance that may increase the risk of cancer, was revealed in the 1940s. The possibility of cancer caused by EC is a health concern [1]. Some studies have indicated that approximately 90% of EC is digested to ethanol, ammonia and carbon dioxide in the rodent liver, and subsequently, about 5% of EC is excreted. Cytochrome P-450 may, however, transform a very small portion of EC into ethyl N-hydroxy carbamate (about 0.1%), a-hydroxy ethyl carbamate (about 0.5%), and vinyl carbamate (about 0.5%). Additionally, N-hydroxy carbamate is a cancer-causing substance that can cause DNA to be oxidized and depurinated by producing nitric oxide and oxygen [2]. Studies on animals have demonstrated that exposure to EC increases the risk of tumor development in a variety of organs, including the lungs [3,4], liver [5,6], and blood vessels. The development of animal embryos can also be affected via EC, according to published findings [7]. In light of these results, the International Agency for Research on Cancer (IARC) upgraded EC from Group 2B (“possible carcinogen to humans”) to Group 2A (“probably carcinogenic to humans”) [8].

The intake of alcoholic beverages might cause the EC exposure, which in turn is harmful to human health. Although EC is not allowed to be added or used in food, it is still a naturally produced chemical compound in fermented foods and alcoholic beverages [9,10,11,12]. This is because fermented foods, especially alcoholic beverages, include some EC precursors, such as urea [13,14], citrulline [15], and cyanide [16]. When combined with ethanol in alcoholic beverages, any of these precursors can eventually introduce EC. EC levels in alcoholic beverages have a widely varying range [11,17,18]. Generally speaking, beers had a low EC content usually, while Chinese rice wine, could have an EC content of up to 775.8 μg/L [19]. Worldwide, European countries [20,21,22], Brazil [23,24,25], Korea [26,27,28] and African countries [29,30] have also reported a number of EC levels in alcohol, with the highest values even reaching over 6000 μg/kg [22]. Health Canada [31], Food Standards Australia New Zealand (FSANZ) [32], and the European Food Safety Authority (EFSA) [33] have all established maximum residue levels (MRLs) of 15–1000 μg/L of EC in alcoholic beverages, and the US Food and Drug Administration (FDA) has developed a voluntary regulation for its monitoring of the health risks of EC ingestion owing to alcohol consumption. China, however, consumes a significant amount of alcohol due to its large population, such that the enactment of laws and limits relating to alcoholic beverages in China will have a profound effect on how society and the economy function. Therefore, data, such as risk assessments of hazardous components in various alcoholic beverages, will need to be continuously examined and updated to evaluate whether increased regulation is required.

The China National Center for Food Safety Risk Assessment (CFSA) conducted a national study, namely, the China total diet study (TDS), on the total diet of the Chinese people in 2016–2019, focusing on the country’s general population. In this article, we used the GC-MS determination method to analyse the EC levels in alcoholic beverages [19]. The intake was calculated by multiplying the EC contents by the most recent dietary consumption survey data for the Chinese population in the sixth TDS. The survey covered 85% of the Chinese population and more than 70% of the country’s land area. Compared with the previous study, the current, sixth TDS was conducted in twenty-four provinces in China, obtaining more representative and authentic general population study data. Margin of exposure (MOE) which was recommended by Joint FAO/WHO Expert Committee on Food Additives (JECFA) and EFSA [34,35] was used in the risk assessment of EC exposure. Virtually safe dose (VSD) and T25 of lifetime cancer risk [36,37,38] was also used to evaluate the chronic carcinogenic risk of EC intake in the general Chinese population. This was a breakthrough in assessing the human health risk of EC ingestion from acute toxicity alone. Finally, we conducted a comparative analysis of EC exposure and risk assessment with a worldwide perspective. The abovementioned information is reported with a view to providing data support for the consumers, food safety, quality control staff, and researchers involved in related policies and regulations.

## 2. Materials and Methods

### 2.1. Reagents and Apparatus

Acetonitrile (HPLC-grade) was purchased from Sigma-Aldrich (St. Louis, MO, USA). Other analytical-grade reagents, including n-hexane, ethyl acetate, and diethyl ether, were purchased from J. T. Baker (Center Valley, PA, USA). Sodium chloride (analytical reagent) was obtained from Sinopharm Chemical Reagent Co., Ltd. (Ningbo Road, SH, CN). EC and d_5_-EC were purchased from A chemtek (Worcester, MA, USA), and the purity was above 99%. Oasis Prime@HLB (30 mg) (Waters, Milford, MA, USA) was introduced in the analysis process. A 7890 A gas chromatograph equipped with the 5977 A mass spectrometer (Agilent Technologies, Santa Clara, CA, USA) was used to perform the identification and quantification processes.

### 2.2. Consumption Survey and Sampling of Alcoholic Beverages

With a few modifications to increase survey efficiency and accuracy, the sixth TDS used the same strategies and framework as previous research. All the included provinces were divided into four primary basket marketplaces based on the geographic factors and regional dietary patterns as follows: North 1 (N1), North 2, South 1, and South 2 (S1). Heilongjiang, Jilin, Liaoning, Beijing, Hebei, and Shanxi made up N1, while Ningxia, Hena, Neimenggu, Shaanxi, Gansu, and Qinghai made up N2. The S1 region included Shanghai, Jiangxi, Fujian, Zhejiang, Shandong, and Jiangsu. The S2 region covered Sichuan, Hubei, Guangxi, Guangdong, Guizhou, and Hunan. For dietary survey and sampling, each province (municipality, autonomous area) selected at least two rural and one urban survey point. When a province’s population exceeds 50 million, the number of survey points should be increased to six, with four rural and two urban sites. Individuals and households completed the dietary survey at the same time. Individual respondents for consumption surveys were collected from 30 randomly selected households at each site. The average food consumption of a standard Chinese male (18–45 years old, 63 kg body weight) was collected in 10,000 households (30 households per survey site) over three days of dietary surveys and 24 h of recalls. The survey of two workdays plus one weekend was designed to eliminate the dietary differences between the weekends and working days. This survey mainly records the categories of recipes and their corresponding weights, and the consumption of condiments by household members during the household investigation period. Table 1 provides details on the 261 alcohol samples that were obtained from a total of 100 sampling sites across twenty-four Chinese provinces. The samples from each province represent the major types of alcohol consumed there, and the collection schedule for these samples was based on the results of local questionnaires. To facilitate the assessment of EC intake in these samples, the alcohol collected in each province was mixed according to the consumption survey data. This means that each province that took part in the TDS study ultimately provided only one mixed sample of alcohol for EC content and exposure analysis. An explanation of the sampling procedure in Heilongjiang is as follows: We selected two rural and one urban location in Heilongjiang province at random for sampling sites. One beer and one Baijiu were collected from each sampling site; thus, a total of six single samples were collected. According to this dietary survey, the average intake of the general population in Heilongjiang was 9.41 g of beer and 12.62 g of baijiu. Therefore, the six samples were mixed to form a mixed sample in this ratio. Note: For Baijiu (Liquor), we collected all samples with an alcohol content of 42°. The alcohol content of the samples was lower than 20° after mixing and diluting, which did not affect the extraction efficiency of the pretreatment. Finally, the mixed sample was transported to CFSA via cold transport at −20 °C.

### 2.3. Analytical Procedures

#### 2.3.1. Preparation of Standard Solution

The EC working solution was diluted from the stock solution (100 μg/mL) with hexane at 10 μg/mL. For making the standard calibration curve, six standard solutions were prepared (0.0 μg/L, 10.0 μg/L, 20.0 μg/L, 50.0 μg/L, 100.0 μg/L, and 500.0 μg/L) by diluting the EC working solution (10 μg/mL) with hexane. The standard solution of d_5_-EC was prepared in acetonitrile to reach a final content of 10 μg/mL. Each standard solution was spiked with 200 μg/L d_5_-EC standard solutions.

#### 2.3.2. Analysis Method

With 1 g of alcohol beverage being pipetted into a 10 mL centrifuge tube, the sample was spiked with 40 μL of d_5_-EC standard solution (10 μg/mL) and mixed with 3 mL of distilled water and 1 g of sodium chloride. After ultrasonic extraction for 15 min, the tube was centrifuged at 10,650 g for 10 min. The supernatant was collected, and the residue was extracted with 2 mL of acetonitrile. Then, the supernatant and the 2 mL of acetonitrile were evaporated under nitrogen at 45 °C and then dissolved with 2 mL of n-hexane. The aforementioned solution was added to a diatomite solid-phase extraction column after a 1 min vortex. After 10 min of static time, the analyte was washed with 10 mL of n-hexane, and then eluted with 10 mL of 5% ethyl acetate in diethyl ether. The collected elution was concentrated at 30 °C using a nitrogen flow after being nearly dried with anhydrous sodium sulfate. The supernatant was filtered with a syringe filter of 0.22 µm for the following procedures. Finally, the analyte was diluted to 2 mL of methanol for GC-MS analysis. The injection mode was direct injection with a volume of 1 µL.

#### 2.3.3. Instrumental Analysis

The GC-MS technology detected the target substance under selected ion monitoring (SIM) mode. A VF-WAX column (30 m × 0.39 µm × 0.25 mm i.d.; Agilent, Santa Clara, CA, USA) was applied in this detection. The carrier gas was hydrogen. The injector temperature and detector temperature were set at 250 °C and 270 °C, respectively. The temperature program was as follows: the initial temperature was set at 60 °C, held for 1 min, the sample was linearly heated to 180 °C at 8 °C/min, then to 240 °C at 10 °C/min, and held at this temperature for 5 min. The monitored fragment ions were defined as m/z 62, 74, 89 for EC and m/z 64, 76, 94 for d_5_-EC. The EC and d_5_-EC were identified by comparing the area ratio of respective characteristic ions and relative retention time (±5%). The ion ratios should be consistent with the calibration standards analyzed simultaneously. EC content was quantified by calibration curves obtained from the peak area ratios of EC/d_5_-EC (m/z 62 and m/z 64, respectively).

### 2.4. Quality Control of the Analysis

Linearity, the limit of detection (LOD), and the limit of quantification (LOQ) were calculated by introducing the dietary samples free of EC, which were prepared as depicted above. The standard curve was based on the concentration of each point of the working solution as the horizontal coordinate, and the ratio of the response of each concentration and the response of isotopic internal standard d_5_-EC as the vertical coordinate. The LOD and LOQ were calculated as follows: the standard solution was gradually diluted and added to the blank samples for a relative standard deviation (RSD) of about 30%, and then the lowest content was utilized as the LOD; for LOQ, the RSD value was required to be about 15% [39]. Using a mixed wine sample’s weight of 0.998 kg/L as the basis, LOD and LOQ were converted from μg/L to μg/kg. Methods of trueness assessment were carried out using spiked recovery experiments. Intra-day repeatability and inter-day reproducibility were assessed using the RSD parameters of six dietary samples for one day and six successive days (n = 6), respectively.

### 2.5. Estimated Dietary Intake of Ethyl Carbamate

The estimated daily intake (EDI) of EC was calculated by multiplying the mean EC concentrations in alcoholic beverages by the corresponding food consumption. The applied mean weight for the Chinese population was calculated to be 63 kg, which originated from the China Health and Nutrition Survey 2010–2013 (CHNS 2010–2013) [40]. CHNS was a nutritional survey of the whole Chinese population conducted by the Chinese Center for Disease Control and Prevention (CCDC). Over 69,000 people participated in this survey in 2010–2013. The average weight of 63 kg was calculated from the weight of all healthy participants within the ages of 18–45 who could perform general physical work. In this article, 1/2 LOD (middle bound, MB) was introduced for the exposure evaluation of the EC content below the LOD [41]. Furthermore, the EDIs for the twenty-four provinces as well as the national average EDI were calculated to evaluate the intake risks in different parts of China.

### 2.6. Margin of Exposure Approach

The MOE approach has been applied for the risk assessment of genotoxic and carcinogenic chemicals, such as EC [18,19,23,24,25,26,27,28,29,30]. The MOE was calculated as the ratio of the benchmark dose lower confidence limit of 10% (BMDL_10_) to the estimated dietary intake of EC. The BMDL_10_ is 0.3 mg/kg bw/d according to the JECFA reference in 2006 [34]. The following safety concerns are considered public health risks based on the assessed MOE values: an MOE value of 10,000 indicates possible concern, an MOE value of 10,000–100,000 indicates minimal concern, and an MOE value of more than 100,000 indicates negligible concern [35].

### 2.7. Cancer Risk Approach

Regarding the lifetime cancer risk of EC in humans, the T25 method, one single-dose-response point extrapolation, was adopted in this study. The EU used T25 to specify exact concentration limits for carcinogens [35]. The T25 value was the chronic dose rate that, after adjusting for spontaneous incidence, would cause 25% of the animals to develop tumors at a particular tissue site during that species’ typical lifespan. Additionally, the T25 was divided by the appropriate scaling factor for interspecies dose scaling based on comparative metabolic rates to yield the matching human dosage descriptor, HT25. The human dosage was then computed from the available exposure data. Then, using linear extrapolation, the relevant human lifetime cancer risk was calculated by dividing the exposure dose by the coefficient (HT25/0.25) [35,36,37,38].

Additionally, another lifetime cancer risk level of 1/10^6^ was frequently seen as providing adequate protection, according to the linear-at-low-dose level, which was recommended by the FDA, using the no-threshold method of estimating cancer risk throughout a person’s lifetime. It was determined that 20 to 80 ng/kg bw/d was a VSD [38]. The daily burden under standard eating practices without alcohol is about 20 ng/kg bw/d [38]. When excess cancer risk was calculated, the VSD of 20 ng/kg bw/d and a lifetime risk level of 1/10^6^ were adopted for cancer potency calculation. Additionally, excess cancer risk was equal to 20 ng/kg bw/d (VSD) multiplied by the average daily intake of each province and then divided by 10^6^.

## 3. Results and Discussion

### 3.1. Method Parameter Validation

The generated calibration curves’ correlation coefficients (R^2^) were 0.9999. The LOD and LOQ of this determination method were 2.0 μg/kg and 5.0 μg/kg, respectively. We performed LOD and LOQ analyses with matrix spiking in this article. At the three spiking levels (5 μg/kg, 15 μg/kg, and 25 μg/kg), the outcomes of the recovery studies were 89%, 99%, and 105%, with RSDs of 2.2%, 3.8%, and 4.5%, respectively. For each spiking level, six replicate determinations were made. The intra-day (n = 6) and inter-day (n = 6, 3 days) recovery rates varied from 95.0% to 116.5% and 96.7% to 105.1%, respectively, whereas the RSD varied from 3.3% to 12.5% and 5.3% to 9.7%. As seen in Figure 1, the mass spectrometry peak duration for EC was 13.65 min. In terms of selectivity of detection, a peak with an m/z of 62 at the outgoing peak of d_5_-EC, which was obtained from the ionization of d_5_-EC, can be observed in (A) and (B) of Figure 1. Figure 2 shows the specific reason we speculate for the formation of the m/z 62 peak at d_5_-EC. EC was gradually ionized into charged ions A, B, and C by the ionization of the ion source. In the case of A and B, the ionization did not require the introduction of new H^+^, but in the case of C, the O atom, where linked to the α-carbon, was attacked by new H^+^. This H^+^ might come from the element deuterium itself or hydrogen in the natural environment such that m/z 62 appears at the time of peak emergence at d_5_-EC. It was fortunate that the m/z 62 peak of d_5_-EC was baseline separated from the m/z 62 peaks of EC, as shown in (B) of Figure 1. Therefore, the m/z 62 peaks of EC did not affect the accurate quantification of EC. 

### 3.2. Occurrence of EC in Chinese Dietary Alcohol Beverages

In this determination, a wide range of EC content was found across twenty-four provinces in China. As shown in Figure 3, nationwide, the mean EC content in alcoholic beverages was 10.4 μg/kg. The mean EC content of the spirits in S1 and S2 was higher than in N1 and N2. EC was not detected in the alcoholic beverages of Beijing, Shanghai, and Gansu, which was represented as 1.0 μg/kg by 1/2 LOD. The highest maximum value was 33.8 μg/kg in Guizhou.

From the perspective of risk control, limitations on the EC levels of alcoholic beverages have been introduced in many countries around the world. The MRLs for EC vary among the different alcoholic beverages, and overall, the values of MRLs increase with higher alcohol content [18]. This is because ethanol provides sufficient reaction conditions for the formation of EC [10]. However, as we can see from Section 2.2, the alcoholic beverages of TDS have a specific sample preparation. This sample contained different proportions of beer, baijiu, rice wine, and red wine. The proportion of the mixed sample was determined for each province based on the daily consumption of alcohol by the local population. Since beers with almost no EC content have dominated the daily consumption of the general Chinese population, the level of EC content in the mixed samples must have been diluted. Meanwhile, during the sample collection, we did not exclusively focus on specific high-content samples, but selected sampling sites and samples according to the actual local situation at random. This was more realistic for reflecting people’s daily alcoholic beverage consumption behavior. It is worth noting that the current average daily consumption of alcoholic beverages by the Chinese population is 10.4 µg/kg, which is quite low compared to the MRLs of 15–1000 µg/L [33].

### 3.3. Consumption of EC from Alcoholic Beverages in the Chinese General Population

Alcohol consumption is another important factor influencing the risk assessment, in addition to the EC contents. Table 2 provides statistics on alcohol consumption across the twenty-four provinces. Shanxi had the lowest value of 0.3 g/d, while Shandong had the highest value of 113.6 g/d. According to this consumption survey, the average daily alcohol consumption for adults in twenty-four provinces was 20.1 g/d or 7336.5 g/y. Based on a volume of 0.998 L for 1 kg of alcohol, the consumption of 7336.5 g/y was projected to be 7.32 L/y, which was more in line with the WHO’s stated estimate in 2019 of 6.05 L of alcohol consumption by Chinese adults aged 15 or older [42]. Nationwide, the respective values for Media, P25, P75, and P95 were 14.3 g/d, 3.3 g/d, 24.0 g/d, and 59.0 g/d. As shown in Table 2, there was a noticeable decrease in overall consumption when compared to the fifth TDS. Jiangxi, for instance, experienced a 70% decline from 74.8 g/d to 21.6 g/d. The factors behind the decline in consumption were varied. It was interesting to note that the effects of epidemics such as COVID-19 could all correlate with the decline in alcohol consumption [43,44].

Figure 4 shows that there were clear differences in the consumption of alcoholic beverages among the general population in China. The intake in the S1 and S2 regions in the south was nearly three times higher than in the N1 and N2 regions. This almost predicted that the risk of EC intake in the general population would be higher in southern than in northern China. This was because the two influencing factors of EC exposure, content level and consumption level, were both higher in the southern provinces than in the north. There were significant differences among provinces regarding alcohol consumption. An analysis of geographical factors showed that beer was the dominant product consumed in most Chinese provinces. However, Qinghai, where highland barley wine (4.7 g/d) is made from a plant unique to the Tibetan plateau, dominates local alcohol consumption. Alcohol consumption diversity dictated that it was a more reasonable strategy to apply a point assessment for EC exposure in the general Chinese population.

### 3.4. Estimated Daily Intakes of Alcoholic Beverages in the Chinese General Population

For risk assessment, EDIs are essential. According to Table 2, the EDIs ranged from 0.001 to 24.56 ng/kg bw/d. The mean was 3.23 ng/kg bw/d and was calculated by multiplying the average consumption and average content and dividing by the standard Chinese male’s 63 kg body weight.

Table 2 demonstrates that only four provinces—Heilongjiang, Liaoning, Beijing, and Ningxia—showed a minor increasing trend in comparison to the fifth TDS. Beijing was found to have the largest gain of 110%, rising from 0.20 ng/kg bw/d to 0.44 ng/kg bw/d. However, virtually all of the EDIs in the remaining provinces were much lower. The largest decrease, with a rate of 97.4% from 13.99 to 0.36 ng/kg bw/day, was observed in Shanghai.

The geographical distribution of EDI in China is characterized in Figure 5. The general population intake of EC is higher in the eastern coastal provinces of China (Shandong, Jiangsu, and Zhejiang) than in other provinces. Zhejiang is the region with the highest EC exposure in the whole country. It is worth noting that Chen D W et al. [19] previously reported on the high EC content of Zhejiang rice wine. A range of ND to 691.4 g/L was obtained after 890 samples of rice wine were detected in his report. He also [19] examined the relationship between rice wine’s high EC level and storage period and discovered a favorable association. This might be caused by the rice wine’s high urea concentration and its reaction with ethanol. The presence of rice wine might lead to the increase in EDIs.

### 3.5. Margin of Exposure

According to Table 3, the MOE values for EC intake through alcohol in the general Chinese population varied from 1.2 × 10^4^ to 3.3 × 10^8^. The country’s average was 9.8 × 10^4^. The risk of EC intake through alcohol consumption decreases as MOE increases. The effect of EC on the health risk from daily dietary alcohol intake in the general population in China does not need to cause particular concern. The risk of EC in alcoholic beverages drunk by the general population in China dropped after a five-year break, and the national average value of MOE increased from 8.5 × 10^4^ in the fifth TDS [19] to 9.8 × 10^4^ in the sixth TDS.

Numerous studies in China have evaluated the risks associated with various alcoholic beverages. In a risk analysis of 2039 batches of baijiu across China, Guan T et al. [18] found that the mean MOE for strong and Jiang flavor baijiu was 4098 and 8036, respectively. The MOE for all varieties of baijiu was 7500 (mean). The average amount of alcohol consumed daily in China is 0.04 L [45], according to data from the industry. A risk analysis of EC exposure through Chinese rice wine was also carried out by Chen D. et al. [19], who indicated that the health risks of rice wine were of concern after obtaining MOEs of 726 to 3913 (mean). Meanwhile, the international investigation of the dangers of EC for particular wine products and demographics has been concentrated in Korea and Brazil. The MOE was usually always more than 10,000 despite Korea’s continuous publication of EC intake and risk studies for numerous alcoholic beverages in recent years. The MOE for a recent risk assessment of EC intake from alcoholic beverages for the Korean population was 14,874 [28], as opposed to Brazil [24], which had an extremely low MOE of 5085 for cachaça and 2305 for other beverages.

### 3.6. Lifetime Cancer Risk

According to Table 3, the lifetime cancer risk range for twenty-four Chinese provinces was between 4.6 × 10^−11^ and 1.2 × 10^−6^, with the highest and lowest values occurring in Hebei and Zhejiang, respectively. The national mean was 1.5 × 10^−7^, which can be translated as a cancer incidence rate in China’s general population of 1.5 per 10 million persons due to EC intake through alcohol drinking. According to this finding, China’s general population does not consume enough alcohol to increase their risk of developing cancer.

In the meantime, the T25 approach [36,46], a different assessment procedure, was employed to determine the carcinogenicity of EC exposure. The results are similarly displayed in Table 3, with lifetime cancer risk ranging from 2.6 × 10^−11^ to 6.9 × 10^−7^ for twenty-four provinces with an average of 8.6 × 10^−8^ nationwide. H25/0.25 was produced by using 1.0 mg/kg bw/d [37] of the T25 for EC as the starting point and assuming a body weight of 63 kg for the conversion. The EDIs were then divided by H25/0.25 per kilogram intake of the standard Chinese male to arrive at the result. The ultimate lifetime cancer risk calculated using the T25 method and the VSD approach had the same order of magnitude, as shown in Table 3. However, T25 had a greater cancer risk. The extrapolation of the three datasets was not linear, as is shown when we contrast the doses of VSD, BMDL_10_, and T25. The findings of either risk assessment indicate that the average Chinese population’s EC intake of alcohol is at a risk level that can be easily managed.

## 4. Conclusions

In this article, the group determined the EC levels of daily alcoholic beverages in the general Chinese population with the support of the platform of the sixth TDS. A risk analysis of EC intake in conjunction with the national alcohol consumption survey was conducted. The point assessment approach used in this study, covering 85% of China’s population and more than 70% of the country’s land area, reflected the most realistic EC health risk of daily alcohol intake in the general Chinese population. The results show that the EC levels and risk assessment should not cause excessive concern, although China has not yet set MRLs for EC in alcoholic beverages. As the country with the world’s highest levels of alcohol production and consumption, this study could provide data to support the development of alcohol policies and regulations in China. In addition, we present data supporting implications for related studies on food safety, food quality, food control, and epidemiology. 

The risk assessment model had the advantage of being economical, efficient, and convenient to derive the health risk of EC exposure in the general population within a short timeframe. However, it also had some shortcomings and limitations. Firstly, this survey only targeted the main consumption varieties and did not capture all of the Chinese alcoholic beverages. Secondly, the participation of the general population did not accurately differentiate between high-risk groups such as alcoholics. Therefore, further precise probability assessments could be conducted in subsequent studies to address these shortcomings and obtain more targeted data.

## Figures and Tables

**Figure 1 foods-12-03129-f001:**
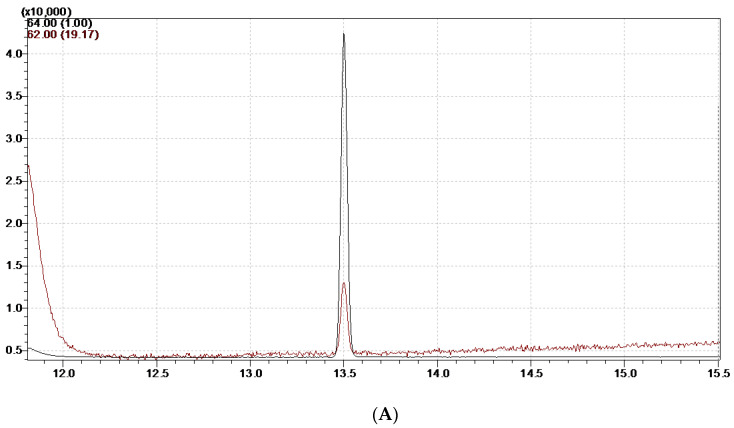
The select ion mode (SIM) diagram for EC (m/z = 62) and d_5_-EC (m/z = 64). (**A**) Blank instrument system; (**B**) standard solution with a concentration of 10 μg/L; (**C**) SIM diagram of a real mixed sample from Jiangxi Province at a concentration of 11.6 10 μg/kg.

**Figure 2 foods-12-03129-f002:**
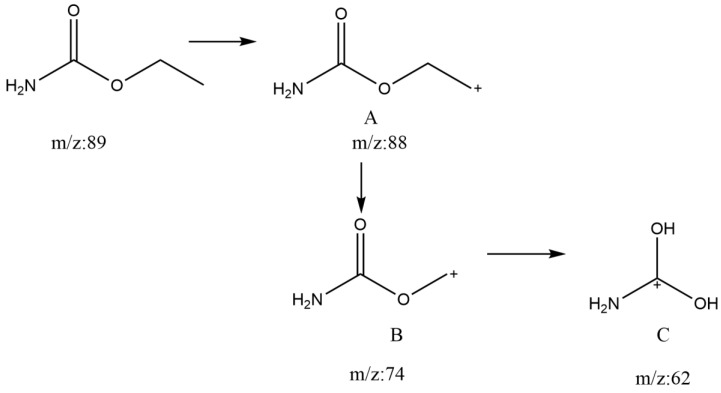
Schematic representation of the ionization process of EC in GC-MS.

**Figure 3 foods-12-03129-f003:**
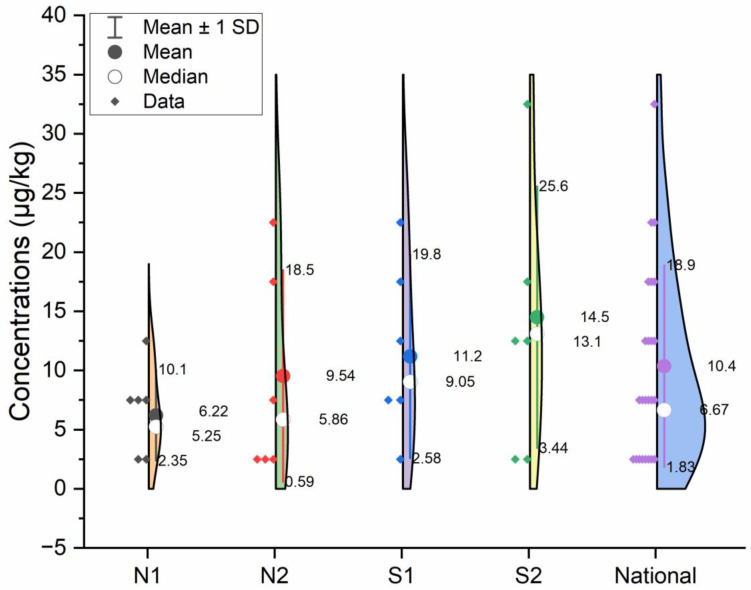
EC content of alcoholic beverages by province in the sixth TDS.

**Figure 4 foods-12-03129-f004:**
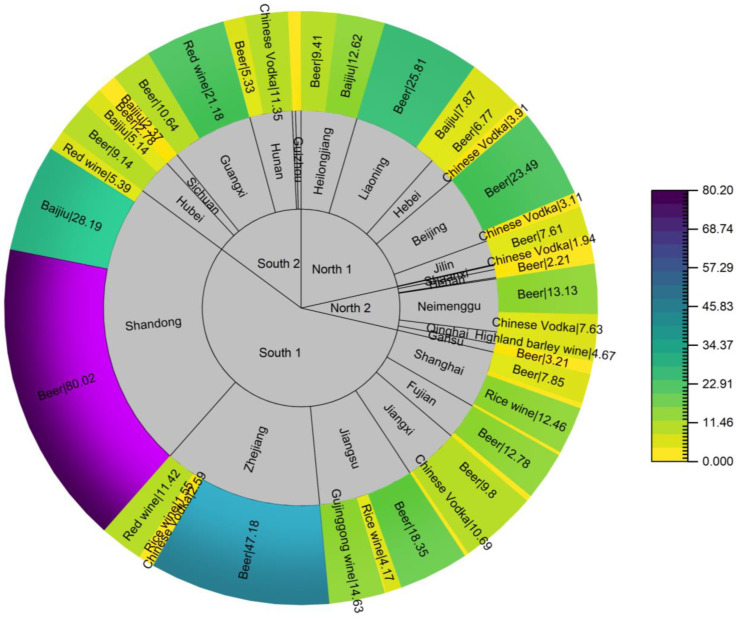
Sunburst graph of alcohol beverage consumption in the Chinese general population from the 6th TDS (g/d).

**Figure 5 foods-12-03129-f005:**
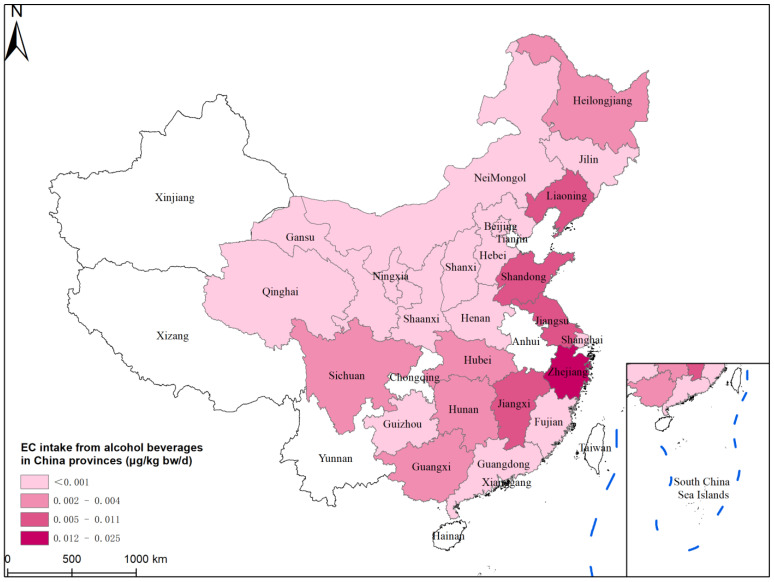
EC intake of the Chinese general population from alcoholic beverages in Chinese provinces.

**Table 1 foods-12-03129-t001:** Sampling sites and information for alcoholic beverages for the general population in China.

Provinces	Type and Number of Sampling Sites	Sample Information for Each Sampling Site
Jiangsu	4 rural + 2 urban	Gujinggong Wine; Rice Wine; Red Wine
Hunan	4 rural + 2 urban	Beer; Baijiu
Shanghai	2 rural + 1 urban	Rice Wine; Beer; Baijiu; Wine
Jiangxi	2 rural + 2 urban	Beer; Rice Wine; Chinese Vodka
Zhejiang	4 rural + 2 urban	Beer; Rice Wine; Red Wine; Chinese Vodka
Gansu	2 rural + 1 urban	Beer; Baijiu
Guizhou	2 rural + 1 urban	Beer; Rice Wine; Baijiu
Nei Mongolia	2 rural + 1 urban	Beer; Chinese Vodka
Hebei	4 rural + 2 urban	Beer; Baijiu
Beijing	2 rural + 1 urban	Beer; Chinese Vodka; Red Wine
Fujian	2 rural + 1 urban	Beer; Baijiu; Daqu Liquor
Jilin	2 rural + 1 urban	Beer; Chinese Vodka
Heilongjiang	2 rural + 1 urban	Beer; Baijiu
Shaanxi	2 rural + 1 urban	Beer; Chinese Vodka
Liaoning	2 rural + 1 urban	Beer; Baijiu
Henan	4 rural + 2 urban	Beer; Chinese Vodka; Wine
Guangdong	4 rural + 2 urban	Beer; Red Wine; Rice Wine; Brandy
Guangxi	2 rural + 1 urban	Beer; Baijiu; Red Wine
Hubei	4 rural + 2 urban	Beer; Baijiu
Qinghai	2 rural + 1 urban	Highland Barley Wine
Shandong	4 rural + 2 urban	Beer; Red Wine; Baijiu
Ningxia	2 rural + 1 urban	Chinese Vodka
Shanxi	2 rural + 1 urban	Baijiu
Sichuan	4 rural + 2 urban	Beer; Wine; Chinese Vodka

**Table 2 foods-12-03129-t002:** The consumptions and estimated daily intake of EC in alcoholic beverages from the TDS.

Provinces	Consumption (g/d)		Estimated Daily Intake (ng/kg bw/d)	
	The 5th TDS	The 6th TDS	The 5th TDS	The 6th TDS
Heilongjiang	13.3	22.0	1.65	1.78
Liaoning	45.5	33.7	5.63	6.45
Hebei	45.2	10.7	5.81	0.001
Beijing	12.4	27.4	0.20	0.44
Jilin	/	9.6	/	0.69
Shanxi	1.9	0.3	0.14	0.04
Shaanxi	/	1.7	/	0.44
Henan	1.0	3.2	0.42	0.34
Ningxia	0.5	0.5	0.15	0.18
Neimenggu	/	20.8	/	1.23
Qinghai	/	4.7	/	0.37
Gansu	/	3.3	/	0.05
Shanghai	25.1	22.5	13.99	0.36
Fujian	45.3	14.3	7.27	1.48
Jiangxi	74.8	21.6	28.50	5.93
Jiangsu	37.5	37.5	9.46	6.88
Zhejiang	69.2	62.7	40.51	24.56
Shandong	/	113.6	/	6.03
Hubei	19.8	14.3	9.71	2.75
Sichuan	14.7	5.2	3.99	1.55
Guangxi	4.2	32.0	2.74	2.08
Hunan	11.3	16.7	5.35	3.73
Guangdong	/	1.3	/	0.08
Guizhou	/	2.2	/	1.17
Mean	26.4	20.1	8.30	3.23
Media	17.3	14.3	/	/
P25	9.5	3.3	/	/
P75	45.2	24.0	/	/
P95	70.6	59.0	/	/

/: no data.

**Table 3 foods-12-03129-t003:** The MOE and lifetime cancer risk in the Chinese general population.

Provinces	MOE	VSD	T25
Heilongjiang	1.7 × 10^5^	8.9 × 10^−8^	5.0 × 10^−8^
Liaoning	4.7 × 10^4^	3.2 × 10^−7^	1.8 × 10^−7^
Hebei	3.3 × 10^8^	4.6 × 10^−11^	2.6 × 10^−11^
Beijing	6.9 × 10^5^	2.2 × 10^−8^	1.2 × 10^−8^
Jilin	4.3 × 10^5^	3.5 × 10^−8^	2.0 × 10^−8^
Shanxi	6.9 × 10^6^	2.2 × 10^−9^	1.2 × 10^−9^
Shaanxi	6.9 × 10^5^	2.2 × 10^−8^	1.2 × 10^−8^
Henan	8.8 × 10^5^	1.7 × 10^−8^	9.6 × 10^−9^
Ningxia	1.7 × 10^6^	8.8 × 10^−9^	4.9 × 10^−9^
Neimenggu	2.4 × 10^5^	6.1 × 10^−8^	3.4 × 10^−8^
Qinghai	8.2 × 10^5^	1.8 × 10^−8^	1.0 × 10^−8^
Gansu	5.7 × 10^6^	2.6 × 10^−9^	1.5 × 10^−9^
Shanghai	8.4 × 10^5^	1.8 × 10^−8^	1.0 × 10^−8^
Fujian	2.0 × 10^5^	7.4 × 10^−8^	4.2 × 10^−8^
Jiangxi	5.1 × 10^4^	3.0 × 10^−7^	1.7 × 10^−7^
Jiangsu	4.4 × 10^4^	3.4 × 10^−7^	1.9 × 10^−7^
Zhejiang	1.2 × 10^4^	1.2 × 10^−6^	6.9 × 10^−7^
Shandong	2.7 × 10^4^	5.5 × 10^−7^	3.1 × 10^−7^
Hubei	1.1 × 10^5^	1.4 × 10^−7^	7.7 × 10^−8^
Sichuan	1.9 × 10^5^	7.7 × 10^−8^	4.4 × 10^−8^
Guangxi	1.4 × 10^5^	1.0 × 10^−7^	5.9 × 10^−8^
Hunan	8.0 × 10^4^	1.9 × 10^−7^	1.0 × 10^−7^
Guangdong	3.6 × 10^6^	4.2 × 10^−9^	2.4 × 10^−9^
Guizhou	2.6 × 10^5^	5.8 × 10^−8^	3.3 × 10^−8^
mean	9.8 × 10^4^	1.5 × 10^−7^	8.6 × 10^−8^

## Data Availability

The raw data supporting the conclusions of this article will be made available by the authors without undue reservation.

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
