# Peer review of "A Nationwide Survey and Risk Assessment of Ethyl Carbamate Exposure Due to Daily Intake of Alcoholic Beverages in the Chinese General Population"

_foods, 2023, doi:10.3390/foods12163129_

Round 1
Reviewer 1 Report
The aim of the paper it’s not clear and should be defined
Line 78: the dietary consumption survey should include a reference
Table 1 The amount of alcohol (%) should be described for each sample
In the section 2.4 and 2.5 all the statements should be supported by different references. (Line 175-176-177)
Line 192: the method should be supported by a reference.
The findings of the work may appear to be trivial if are not well discussed in the context of the extant body of literature. Authors can build a discussion to help readers to follow the logical development of the manuscript. The discussion presented is scare and should be included in the sections 3.1; 3.2; 3.3 and enrich in the section 3.4; 3.5.
Conclusion should not be a summary of your study or an extension of the discussion of results. The presented work has some merits, but at the same time, the provided future research directions appear to be trivial.
Author Response
Dear reviewers
Thank you for your suggestions.
I have revised all your comments accordingly and marked them in red in the reworked manuscript. Specific instructions are given below:
1. The aim of the paper it’s not clear and should be defined.
This issue is more of an overarching statement, and I have made changes throughout the text and have highlighted them in red.
2. line 78: the dietary consumption survey should include a reference.
As this is a national consumption survey organized by our center, no literature citation has been made. I have rewritten the sentence as follows: The intake was calculated by multiplying the EC contents by the most recent dietary consumption survey data for the Chinese population in the sixth TDS.
3. Table 1 The amount of alcohol (%) should be described for each sample.
I suspect you may be concerned about the high alcohol content affecting sample extraction. If so, I don't think it's necessary. I made a note on lines 134-136 of the revised draft, which reads: Note: For Baijiu (Liquor) we collected all samples with an alcohol content of 42°, the alcohol content of the samples was lower than 20° after mixing and diluting, which did not affect the extraction efficiency of the pretreatment.
4. In sections 2.4 and 2.5 all the statements should be supported by different references. (Line 175-176-177).
Literature has been introduced in the revised manuscript.
5. Line 192: the method should be supported by a reference.
Literature has been introduced in the revised manuscript.
6. The discussion presented is scary and should be included in sections 3.1; 3.2; 3.3 and enriched in sections 3.4; 3.5.
Sections 3.1 to 3.3 have been rewritten and 3.4-3.5 have been slightly modified. Thank you for your "scared", it made me re-check myself, I did suck the first time, and I hope I didn't scare you this time with my discussion.
7. Conclusion should not be a summary of your study or an extension of the discussion of results. The presented work has some merits, but at the same time, the provided future research directions appear to be trivial.
The conclusion has been rewritten and is completely different. Please review it again.
Please don't hesitate to let me know if you have any questions or suggestions. Thank you for your time and effort on this article.
Kind regards!
Reviewer 2 Report
Dear authors,
I have reviewed your manuscript titled "Assessment of Ethyl Carbamate Levels in Chinese Dietary Alcohol Beverages: Occurrence, Intake, and Carcinogenic Risk" and find it to be a valuable contribution to the field. Your study provides important insights into the occurrence, intake, and carcinogenic risk associated with ethyl carbamate (EC) in Chinese dietary alcohol beverages.
The results demonstrate variations in EC levels across the twenty-four provinces, highlighting the need for regional monitoring. The margin of exposure analysis indicates that the average Chinese person's EC intake through alcohol consumption is at a manageable level. These findings contribute to our understanding of EC intake and its associated risks in the Chinese general population.
Overall, your study adds to the existing knowledge on EC levels, intake, and carcinogenic risk in Chinese dietary alcohol beverages. This research has implications for risk assessors, policymakers, and researchers interested in food safety.
Author Response
Dear reviewers:
Thank you for your approval.
I have revised my manuscript. We hope to do better in the future and look forward to your valuable comments and suggestions on our follow-up work.
Thank you for reviewing and supporting this paper.
Reviewer 3 Report
This manuscript describes a methodology to assess the risk of ethyl carbamate associated with the daily intake of alcoholic beverages in the Chinese population. The document structure is correct. However, I have several comments for the whole paper, which must be taken into account to improve the article.
- The first time that an acronym is mentioned, it should be fully written. For example, GC-MS, RSD.
- Keywords: ethyl carbamate is missing.
- Line 57: indicate the MRL of EC in alcoholic beverages.
- The authors speak of a previous edition (lines 75-76), a 4th TDS (lines 69-70) and a 5th DTS (line 231), but do not include the reference. Apparently, this work does not seem new with respect to that previous study that is mentioned, the study sample was simply expanded (from 16 provinces to 24). The authors must emphasize the differences with respect to this previous study, showing the strengths.
- Lines 76-78: information must be completed. For example, indicate that the EC content levels were determined on a mixed sample of alcoholic beverages by GC-MS. Indicate the reference of the survey in 2019.
- Authors should always use the original reference. Lines 79-80: missing the reference of the MOE method of the JECFA.
- Section 2.1: include all the reagents (hexane, sodium chloride, ethyl acetate, diethyl ether, etc.).
- Lines 98-101: Shaanxi is missing in this classification.
- Lines 106-109: explain in more detail what these surveys and recalls consisted of.
- Line 139: change rpm for g units.
- Line 140: indicate the volume of acetonitrile used.
- Lines 141 and 145: did it evaporate to dryness?
- Line 145: the degree symbol is missing.
- Section 2.3.3: indicate the injection mode and the injection volume.
- Section 2.4: indicate how the linearity was calculated.
- What rule did the authors follow to determine the LOQ and LOD? Indicate the reference.
- Lines 182-183: include the reference of these studies (genotoxicity and carcinogenicity of EC).
- Lines 185-189: indicate the reference to assume these values as possible concern minimal concern or negligible concern.
- Lines 192-193: indicate the reference where the EU uses T25 to specify concentration limits for carcinogens.
- Lines 203-204: indicate the reference of the VSD of 20 to 80 ng/kg bw/day.
- Statistical analysis of the results is missing.
- Lines 239-240: 10.1 µg/kg is the average amount of China or the average amount of Liaoning, Ningxia, and Qinghai? This sentence is confusing.
- Line 248: change the number of the figure and include the standard deviations.
- Lines 254-255: indicate the reference of the WHO´s estimation of the alcohol consumption by Chinese adults.
- Line 256: 116.3 g/d does not correspond to 113.6 g/d in table 2. Please, correct.
- Figure 3 is not mentioned in the text. Maybe, it is not essential.
- Line 277: The EDI of 0.001 ng/kg bw/d corresponds to Hebei (Table 2) instead of Shanxi. Please, correct.
- Line 314: indicate the reference for the value of 8.5E+04 (of the 5th DTS?). The MOE of 9.5E+04 does not correspond to 9.8E+04 in table 3.
- Lines 318-319: indicate the reference of the average amount of alcohol consumed daily in China.
- Lines 326-327: indicate the reference of the Korea results.
- References: write the year in bold.
Author Response
Dear reviewers:
Thank you very much for reviewing my manuscript so meticulously. There are no words to express our gratitude. I will respond to each of your comments below.
- The first time that an acronym is mentioned, it should be fully written. For example, GC-MS, RSD.
I have gone through all the acronyms and made sure to use the full name on the first occurrence. Specifically in the new manuscript in line 20 for GC-MS and line 182 for RSD.
- Keywords: ethyl carbamate is missing.
It has been added.
- Line 57: indicate the MRL of EC in alcoholic beverages.
It has been noted that in line 60 of the new manuscript, for 15-1000 μg/L.
- The authors speak of a previous edition (lines 75-76), a 4th TDS (lines 69-70) and a 5th DTS (line 231), but do not include the reference. Apparently, this work does not seem new with respect to that previous study that is mentioned, the study sample was simply expanded (from 16 provinces to 24). The authors must emphasize the differences with respect to this previous study, showing the strengths.
In the last paragraph of the introduction, the changes have been made following your comments and have been marked in red.
- Lines 76-78: information must be completed. For example, indicate that the EC content levels were determined on a mixed sample of alcoholic beverages by GC-MS. Indicate the reference of the survey in 2019.
For information completion, I have indicated that the EC content levels were determined on a mixed sample of alcoholic beverages by GC-MS in lines 71-73.
For literature citation, as this is a national consumption survey organized by our center, no literature citation has been made. I have rewritten the sentence as follows: The intake was calculated by multiplying the EC contents by the most recent dietary consumption survey data for the Chinese population in the sixth TDS.
- Authors should always use the original reference. Lines 79-80: missing the reference of the MOE method of the JECFA.
It has been added.
- Section 2.1: include all the reagents (hexane, sodium chloride, ethyl acetate, diethyl ether, etc.).
Changes have been made in lines 91-95 of the revised draft.
- Lines 98-101: Shaanxi is missing in this classification.
It has been added in line 106.
- Lines 106-109: explain in more detail what these surveys and recalls consisted of.
I have explained this in lines 118-120 of the article, as follows: This survey mainly records the categories of recipes and their corresponding weights, and the consumption of condiments by household members during the household investigation period.
- Line 139: change rpm for g units.
Already been amended, in line 151.
- Line 140: indicate the volume of acetonitrile used.
Already been amended, in line 152.
- Lines 141 and 145: did it evaporate to dryness?
nearly dried, in line 158.
- Line 145: the degree symbol is missing.
It has been added in line 158.
- Section 2.3.3: indicate the injection mode and the injection volume.
An addition was made in lines 160-161 of the revised draft, which reads: The injection mode was direct injection with a volume of 1 µL.
- Section 2.4: indicate how the linearity was calculated.
An addition was made to lines 178-181 of the revised draft, as follows: The standard curve of this method was based on the concentration of each point of the working solution as the horizontal coordinate and the vertical coordinate was the ratio of the response of each concentration and the response of isotopic internal standard d5-EC.
- What rule did the authors follow to determine the LOQ and LOD? Indicate the reference.
Reference is applied in line 184 of the revised draft.
- Lines 182-183: include the reference of these studies (genotoxicity and carcinogenicity of EC).
As you requested, I have made notes and citations to the corresponding literature in almost every sentence in section 2.6.
- Lines 185-189: indicate the reference to assume these values as possible concern minimal concern or negligible concern.
As you requested, I have made notes and citations to the corresponding literature in almost every sentence in section 2.6.
- Lines 192-193: indicate the reference where the EU uses T25 to specify concentration limits for carcinogens.
The desired references have been cited in line 214 of the article.
- Lines 203-204: indicate the reference of the VSD of 20 to 80 ng/kg bw/day.
The desired references have been cited in line 225 of the article.
- Statistical analysis of the results is missing.
The methods of statistical analyses have been added in lines 227-230 of the article.
- Lines 239-240: 10.1 µg/kg is the average amount of China or the average amount of Liaoning, Ningxia, and Qinghai? This sentence is confusing.
I have rewritten the 3.2 section and the confusing sentence no longer exists.
- Line 248: change the number of the figure and include the standard deviations.
I have redrawn Figure 1, the original has been replaced. This time we will focus only on the results of the sixth TDS test.
- Lines 254-255: indicate the reference of the WHO´s estimation of the alcohol consumption by Chinese adults.
Corresponding literature has been introduced in line 303 of the article.
- Line 256: 116.3 g/d does not correspond to 113.6 g/d in Table 2. Please, correct.
Already harmonized at 113.6 g/d.
- Figure 3 is not mentioned in the text. Maybe, it is not essential.
Figure 4 is needed for an intuitive data visualization (changed from Figure 3 to Figure 4 in the newly revised manuscript). we had mentioned Figure 4 on line 312 of the article.
- Line 277: The EDI of 0.001 ng/kg bw/d corresponds to Hebei (Table 2) instead of Shanxi. Please, correct.
It had been amended to read Hebei, in line 333.
- Line 314: indicate the reference for the value of 8.5E+04 (of the 5th DTS?). The MOE of 9.5E+04 does not correspond to 9.8E+04 in table 3.
The corresponding reference was already cited and the entire sentence was modified to read: The risk of EC in alcoholic beverages drunk by the general population in China dropped after a five-year break and the national average of MOE increased from 8.5E+04 of the fifth TDS [19] to 9.8E+04 of the sixth TDS. (lines 370-371)
- Lines 318-319: indicate the reference of the average amount of alcohol consumed daily in China.
The literature had been introduced in line 376.
- Lines 326-327: indicate the reference of the Korea results.
The literature had been introduced in lines 385-386.
31. References: write the year in bold.
The years of all references had been written in bold.
Please don't hesitate to let me know if you have any questions or suggestions. Thank you for your time and effort on this article.
Kind regards!
Reviewer 4 Report
The article covers the Aims & Scope of the Journal reporting the results of a national survey and risk assessment ofethyl carbamate from daily intake of alcoholic beverages in the Chinese general population.
The document is well-structured and the provided information provided are of importance. I recommend the publication of the manuscript considering some minor comments.
Comments to authors:
p.5/Ln.165: Instead for “Recovery analyses”, please refer to “methods trueness assessment”.
p.5/Ln. 176: The authors refer to a calculated bw of 63 kg. How was this mean weight calculated? Please provide an explanation/description.
Figure 1:
- Did the authors identify or assess what is the second peak exists at 14.6 min? Is this is an impurity or an isomer? It seems that you have a similar peak exist in the d5 chromatogram. Please discuss and explain.
- Also it seems that there is another peak at ca.15.5 min for both ions. Please amend and discuss.
- Maybe to provide a full chromatogram and a zoomed (selected) area can be of help for the reader.
Figure 1 caption: Please provide the concentration of EC and d5-EC (200 μg/L as mentioned in section 2.3.1).
The disclaimer on the last page (L.481-483) shall be placed before the list of references.
Author Response
Dear reviewers:
Thank you very much for reviewing my manuscript so meticulously. There are no words to express our gratitude. I will respond to each of your comments below.
- p.5/Ln.165: Instead for “Recovery analyses”, please refer to “methods trueness assessment”.
I have made changes, in line 186 of the returned manuscript. The details are: Methods of trueness assessment were carried out using spiked recoveries experiments.
- p.5/Ln. 176: The authors refer to a calculated bw of 63 kg. How was this mean weight calculated? Please provide an explanation/description.
In lines 191-198, I explained as follows: the applied mean weight for the Chinese population was calculated to be 63 kg (bw) which this data originated from China Health and Nutrition Survey 2010-2013 (CHNS 20102013) [40]. CHNS was a nutritional survey of the whole Chinese population conducted by the Chinese Center for Disease Control and Prevention (CCDC). There were over 69,000 people participated in this survey in 2010-2013. The average weight of 63 kg was calculated from the weight of all healthy participants between the ages of 18-45 who could perform general physical work.
- For an explanation of Figure 1
Firstly, due to the variability of the samples across provinces, some samples have more and some have less interference. But the peak time is fixed. Therefore, for compounds with a large time we had not conducted a qualitative analysis. We found the samples with less matrix interference according to your request and re-plotted them according to the calibrated concentration and local amplification. A new Figure 2 was added to explain why the peak at m/z 62 appeared during the internal standard peak-out time.
- The disclaimer on the last page (L.481-483) shall be placed before the list of references.
It has been modified.
Please don't hesitate to let me know if you have any questions or suggestions. Thank you for your time and effort on this article.
Kind regards!
Round 2
Reviewer 4 Report
The authors have addressed all my comments. I recommend the publication of this research.